# Hemodynamic Response Asymmetry During Motor Imagery in Stroke Patients: A Novel NIRS-BCI Assessment Approach

**DOI:** 10.3390/s25165040

**Published:** 2025-08-14

**Authors:** Mikhail Isaev, Pavel Bobrov, Olesya Mokienko, Irina Fedotova, Roman Lyukmanov, Ekaterina Ikonnikova, Anastasiia Cherkasova, Natalia Suponeva, Michael Piradov, Ksenia Ustinova

**Affiliations:** 1Institute of Higher Nervous Activity and Neurophysiology of the Russian Academy of Sciences, Moscow 117485, Russia; isaev.mikhail@ihna.ru (M.I.); bobrov.pavel@ihna.ru (P.B.); irinfed@mail.ru (I.F.); 2Department of Neuro-Computer Interfaces, Pirogov Russian National Research Medical University, Moscow 117513, Russia; xarisovich@gmail.com; 3Russian Center of Neurology and Neurosciences, Moscow 125310, Russia; ikonnikovaes@list.ru (E.I.); cherka.sova@mail.ru (A.C.); dir@neurology.ru (M.P.); 4Department of Physical Therapy, Central Michigan University, Mount Pleasant, SC 48859, USA; ustin1k@cmich.edu

**Keywords:** human–computer interaction, near-infrared spectroscopy, brain–computer interface, stroke rehabilitation, motor imagery, interhemispheric asymmetry, laterality index, hemodynamic response, biomarkers

## Abstract

**Highlights:**

**What are the main findings?**
A novel task response asymmetry coefficient is introduced for assessing daily dynamics of interhemispheric hemodynamic response asymmetry in post-stroke patients and healthy individuals.The proposed task response asymmetry coefficient could be used for investigating interhemispheric dynamics even in heterogeneous groups of patients.

**What is the implication of the main finding?**
For the patients, the proposed coefficient indicates significant difference between lesioned and intact hemisphere in terms of response to affected and intact hand movement imagery, as well as evident daily dynamics of the asymmetry for people with substantial recovery.Daily dynamics in functional asymmetry in post-stroke patients engaged in various rehabilitation procedures is required.

**Abstract:**

Understanding patterns of interhemispheric asymmetry is crucial for monitoring neuroplastic changes during post-stroke motor rehabilitation. However, conventional laterality indices often pose computational challenges when applied to functional near-infrared spectroscopy (fNIRS) data due to the bidirectional hemodynamic responses. In this study, we analyze fNIRS recordings from 15 post-stroke patients undergoing motor imagery brain–computer interface training across multiple sessions. We compare traditional laterality coefficients with a novel task response asymmetry coefficient (TRAC), which quantifies differential hemispheric involvement during motor imagery tasks. Both indices are calculated for oxygenated and deoxygenated hemoglobin responses using general linear model coefficients, and their day-to-day dynamics are assessed with linear regression. The proposed TRAC demonstrates greater sensitivity than conventional measures, revealing significantly higher oxygenated hemoglobin TRAC values (0.18 ± 0.19 vs. −0.05 ± 0.20, *p* < 0.05) and lower deoxygenated hemoglobin TRAC values (−0.15 ± 0.27 vs. 0.04 ± 0.23, *p* < 0.05) in lesioned compared to intact hemispheres. Among patients who exhibit substantial motor recovery, distinct daily TRAC dynamics were observed, with statistically significant temporal trends. Overall, the novel TRAC metric offers enhanced discrimination of interhemispheric asymmetry patterns and captures temporal neuroplastic changes not detected by conventional indices, providing a more sensitive biomarker for tracking rehabilitation progress in post-stroke brain–computer interface applications.

## 1. Introduction

Understanding the compensatory mechanisms underlying motor function recovery after stroke remains a critical area of research. While structural changes facilitate recovery, functional reorganization is believed to reflect underlying neuroplastic adaptations [1,2]. Among neuroplasticity indicators in unilateral focal brain lesions, indices of interhemispheric asymmetry provide particularly informative metrics. The laterality coefficient (or index) is a key measure, quantifying differential activation between homotopic regions of the affected and intact hemispheres and their relative contributions to motor task execution [2,3]. In healthy individuals performing motor tasks, the hemodynamic response typically demonstrates predominant activation of the primary sensorimotor cortex of the contralateral hemisphere, with less prominent activation in ipsilateral homotopic areas [4,5]. However, when post-stroke patients with unilateral paresis attempt to perform the same tasks with their affected limb, this normal activation pattern becomes disrupted. Specifically, contralateral hemisphere activation diminishes [6,7], while ipsilateral activation increases as a compensatory response, resulting in a shift in the hemispheric activation pattern toward the intact side [4,6].

Studies utilizing electroencephalography (EEG) and functional magnetic resonance imaging (fMRI) demonstrate that, during attempts to move the affected limb, post-stroke patients consistently show significantly reduced laterality index values compared to healthy controls [2,4]. Additionally, there is an inverse relationship between the laterality index and the severity of motor deficits in post-stroke patients [7,8,9]. Longitudinal studies document progressive increases in the laterality index during motor recovery, suggesting adaptive reorganization [4,10,11,12]. This activation of the intact hemisphere is thought to facilitate the recruitment of additional neural resources, compensating for impaired motor function [13].

Asymmetry measures, such as the laterality index, are often derived based on EEG and fMRI data. EEG is extensively used to assess the effectiveness of motor imagery training with brain–computer interfaces (BCIs), allowing real-time monitoring of cortical activity during training sessions [11,12,14,15]. BCIs detect and interpret biological signals—most commonly EEG—to bridge brain activity and computerized systems. Given that brain activity is recorded during each BCI session, this methodology presents unique opportunities for tracking the longitudinal dynamics of asymmetry indices on a session-by-session basis. However, these studies have predominantly restricted their analyses to pre- and post-intervention comparisons.

Near–infrared spectroscopy (NIRS) is an alternative to fMRI for recording brain metabolic activity. NIRS has been successfully incorporated into BCI systems [16,17], though its application in post-stroke rehabilitation through NIRS-BCI paradigms remains limited, with only a few studies published to date [18,19,20,21]. Importantly, none of these studies have monitored longitudinal changes in laterality index during the rehabilitation process. Several recent non-BCI studies have employed NIRS to assess laterality, during actual movement executions, but not during motor imagery, focusing primarily on sensorimotor and other cortical regions in both healthy controls and stroke patients [22,23,24,25,26]. Thus, a significant knowledge gap persists in the field. To our knowledge, no studies have systematically examined the session-by-session evolution of key neurophysiological parameters, including hemodynamic response magnitude during motor imagery and laterality indices, in post-stroke patients undergoing BCI-based rehabilitation. Understanding these temporal dynamics could yield novel insights into neuroplastic reorganization and inform individualized treatment protocols for post-stroke motor recovery.

Calculating the laterality index in functional NIRS (fNIRS) studies can be based on the averaged amplitude of the hemodynamic response or through coefficients derived from the general linear model (GLM) [27]. However, applying conventional laterality formulas to NIRS data presents significant methodological challenges that distinguish it from other neuroimaging modalities. Unlike EEG or fMRI, where computed metrics such as spectral power, rhythm suppression indices, or activation cluster sizes are inherently positive values, NIRS–derived measures including averaged hemodynamic responses and GLM coefficients can assume negative values. This fundamental difference creates computational complexities when applying traditional laterality calculations. These limitations underscore the ongoing need for developing robust methodological approaches for computing interhemispheric asymmetry indices specifically tailored to NIRS data characteristics. However, unlike EEG and fMRI, NIRS-derived indices such as mean hemodynamic amplitude or GLM coefficients may be negative, complicating the application of traditional laterality metrics designed for positive-valued data. This methodological challenge underscores the need for robust, NIRS-specific approaches for quantifying interhemispheric asymmetry. Such methods should accurately capture both the magnitude and directionality of the hemodynamic response, while avoiding the mathematical instabilities of conventional laterality calculations. Addressing this gap is critical for advancing NIRS-based BCI research and post-stroke rehabilitation.

The objectives of this study are twofold: first, to systematically investigate the day-to-day dynamics of the laterality coefficient during NIRS-BCI training; and second, to introduce and validate a novel metric of interhemispheric hemodynamic asymmetry tailored to the unique properties of NIRS biosignals for monitoring motor imagery performance in post-stroke patients. 

## 2. Materials and Methods

### 2.1. Datasets

We analyzed NIRS recordings, obtained from a previously conducted clinical NIRS-BCI study [21,28]. The study involved 15 patients, 9 men and 6 women, all right-handed, with a median age of 58.8 years [25% and 75% quartile range: 49.4–70.0 years]. The median time since stroke onset was 7.0 months [2.0–10.0 months]. All patients had unilateral cortical lesions, with eight lesions located in the left hemisphere and seven in the right hemisphere. At baseline, the median Fugl–Meyer Assessment for Upper Extremity (UE–FMA) score was 47.0 [35.0; 54.0]; the median baseline Action Research Arm Test (ARAT) score was 35.0 [10.0; 44.0] (Table 1).

As the control group, healthy participants from a previous study were selected, in which they imagined movements of the left or right hand to control the NIRS-BCI within a similar experimental paradigm [29]. This group consisted of 8 men and 1 woman, all right-handed, with a median age of 28 years [22; 39] years.

The experimental design and data acquisition procedures for the studies from which the current analysis datasets were obtained are described in the following two subsections. 

### 2.2. Experimental Design

Each subject, wearing an NIRS cap, was seated comfortably in front of a monitor with their hands resting on the armrests or on a table. The screen displayed a black background featuring a central fixation circle surrounded by three gray arrows. These arrows indicated the required tasks: the top arrow represented the rest period, while the left and right arrows corresponded to imagining left- and right-hand movements, respectively. A color change to blue signaled preparation, and a change to green indicated the onset of the corresponding task. Correct task classification was indicated by the circle expanding and turning green, while incorrect classification resulted in the circle shrinking. No feedback was provided during rest or preparation phases. The classification algorithm is described in detail in a previous publication [29].

During the clinical experiments, each patient participated in 1–2 sessions per day, for a total of 7 to 24 sessions over a period of 7 to 15 days. Each session consisted of either 4 or 6 blocks, lasting approximately 9 or 14 min, respectively. Each block included 4 trials, 2 trials of right-hand motor imagery and 2 trials of left-hand motor imagery, presented in random order. Each trial comprised a 17 s rest phase followed by a 17-s motor imagery phase; within each phase, participants had 2 s for preparation and 15 s for task execution. Classification (for both motor imagery and rest) was performed on 1 s data segments with a 250 ms shift window. During the motor imagery phase, feedback was updated according to the classification results. The full study protocol has been described previously [21].

Healthy control subjects participated in experiments for up to 11 days, following a similar block structure. Each session included three blocks. In each phase, participants were given 3 s for preparation and 20 s for task execution. The same BCI classification algorithm was employed, and as in the clinical cohort, no feedback was provided during the rest or preparation phases.

### 2.3. Data Acquisition

Clinical data were recorded using the NIRScout system (NIRx Medizintechnik GmbH, Berlin, Germany) equipped with 8 detectors and 16 light emitters (wavelengths of 760 and 850 nm), of which 14 sources were used. The sampling rate was set to 15.6 Hz. Optodes were positioned over the motor cortex areas with an approximate 3 cm distance between them. Emitters were placed at positions F3, FC5, FC1, C3, CP5, CP1, P3, F4, FC2, FC6, C4, CP2, CP6, and P4, and the detectors were located at FC3, C5, C1, CP3, FC4, C2, C6, and CP4. A total of 28 emitter–detector pairs were selected to form signal recording channels. The recording methodology is documented in [21,28]. The study by Isaev et al. [28] also provides a repository link to the NIRS recordings.

Control data were recorded using the same device, with a sampling rate of 3.9 Hz. Emitters were placed at CCP1h, FCC3h, CCP5h, FFC5h, FFC1h, CPP3h, AFF1, FTT7h, FCC2h, FCC6h, FFC4h, CCP4h, AFF2, CPP6h, CPP4h, and CPP2h, and the detectors at FCC1h, CCP3h, FCC5h, FFC3h, FFC2h, CCP2h, FCC4h, and CCP6h. A total of 33 emitter–detector pairs formed the data channels, 14 of which coincided with those in clinical experiments.

The difference in sampling rates is attributable to different illumination patterns. In an earlier study with healthy subjects, all the NIRS emitters were illuminated sequentially, one at a time, while in clinical experiments the emitters were grouped in four clusters and illuminated simultaneously, based on prior findings of crosstalk absence in this configuration.

### 2.4. Data Processing

All data were processed using Matlab 2023b (MathWorks, inc, USA) and R (R-4.5.0; R Core Team 2025).

### 2.5. Hemodynamic Response Estimation

Raw NIRS intensity data for the specified wavelengths were processed without preliminary filtering. Relative concentrations of oxyhemoglobin (HbO) and deoxyhemoglobin (HbR) were calculated using the modified Beer–Lambert law, as described previously [28]. The obtained concentration values were filtered using a 4th-order zero-phase Chebyshev bandpass filter with cutoff frequencies of 0.005 and 0.09 Hz.

Next, a general linear model (GLM) was constructed for all the recordings, using experimental markers as predictors. Boxcar functions corresponding to periods of left- and right-hand motor imagery movements and task preparation (including rest) were used as regressors. Additional delta-functions modeled responses to color changes in arrows, capturing visual cue onsets for all instructions except preparatory ones. All regressors were convolved with a standard hemodynamic response function. Responses to preparatory instructions highly correlated with the onset responses after the convolution due to their short duration and, therefore, were not included in the model. The resting state served as the baseline level in GLM. The model hemodynamic response function was computed using the SPM12b package tools (https://www.fil.ion.ucl.ac.uk/spm/), employing the spm_hrf function with default parameters. The scanning time was set to 1/Fs, where Fs is the NIRS sampling rate. The GLM coefficients for the corresponding regressors were taken as the quantitative responses to left- or right-hand imaginary movement.

### 2.6. Interhemispheric Hemodynamic Response Asymmetry Metrics

The amplitude of hemodynamic responses, as estimated by the GLM coefficients, was used to calculate two indices of interhemispheric asymmetry, the laterality coefficient (LC) and the task response asymmetry coefficient (TRAC), the latter introduced in the present work.

The laterality coefficient was calculated as(1)$$LC=\frac{R_{h,contra}-R_{h,ipsi}}{\sqrt{2\left(R_{h,contra}^{2}+R_{h,ipsi}^{2}\right)}},$$
where *R_h,contra_* is the amplitude (signed) of the hemodynamic response (i.e., the corresponding GLM coefficient) to imagining the movement of hand h (either affected or intact) in the contralateral hemisphere, and *R_h,ipsi_* is the amplitude of the response in the ipsilateral hemisphere. 

The task response asymmetry coefficient was calculated as (2)$$TRAC=\frac{R_{contra,k}-R_{ipsi,k}}{\sqrt{2\left(R_{contra,k}^{2}+R_{ipsi,k}^{2}\right)}}$$
where *R_contra,k_* is the amplitude of the hemodynamic response on the k-th channel while imagining the movement of the contralateral hand, and *R_ipsi,k_* is the response while imagining the movement of the ipsilateral hand.

Before calculating coefficients for the patient data, we mirrored symmetric channels across the left and right hemispheres in participants with right-sided lesions. This ensured that, for all the patients, channels on the left side corresponded to the lesioned hemisphere and those on the right to the intact hemisphere, allowing for direct comparison of coefficients across the cohort. No channel mirroring was applied for the healthy participants. Coefficients were calculated separately for HbO and HbR responses.

### 2.7. Statistical Analysis

LC and TRAC values for HbO and HbR responses were analyzed separately. For each participant and channel, coefficient values were averaged, and a repeated-measures nonparametric factorial ANOVA with aligned rank transform was performed using the ARTool package in R [30]. For LC, the model included two within-subject factors: hand (affected/intact) and channel (14 symmetric pairs). For TRAC, the factors were hemisphere (lesioned/intact) and channel (14 symmetric pairs). In healthy participants, where lesioned/intact distinctions were not applicable, the LC and TRAC values were analyzed with respect to left and right hands or hemispheres.

Next, for each pair of symmetric channels, the corresponding LC or TRAC values were compared using the Wilcoxon signed-rank test. To account for multiple comparisons, *p*-values were adjusted using the Benjamini–Yekutieli procedure.

To evaluate daily variations, coefficient values were averaged across sessions for each participant, channel, and day. Visual inspection revealed no evidence of nonlinear monotonic trends, justifying the use of linear regression models to examine linear relationships between indices and day number. Separate models were fitted for each coefficient (LC/TRAC), signal type (HbO/HbR), and channel. The statistical significance of regression slopes was assessed with Benjamini–Yekutieli-adjusted *p*-values.

In patients, Pearson correlation analysis was performed to assess the relationship between clinical outcomes and interhemispheric asymmetry metrics as well as their corresponding regression slopes. Baseline ARAT scores and relative post-therapy ARAT improvement were used for the analysis.

## 3. Results

### 3.1. LC and TRAC Numerical Stability

According to Equation (1), the LC quantified the difference in responses to a specific task between homotopic areas of the lesioned and intact hemispheres, serving as a direct measure of interhemispheric asymmetry. Calculating the LC required selecting a task and a pair of homotopic channels.

As defined by Equation (2), the TRAC measured the difference in responses within a selected area to two different tasks, thus allowing comparison of the area’s involvement in each task. Calculating TRAC required selecting a channel and a pair of tasks. Although TRAC itself did not directly measure interhemispheric asymmetry, this could be inferred by comparing TRAC values between homotopic areas. It should be noted that, in terms of numbers of channels and samples per channel, both indices used the same amount of data: assuming equivalent task durations, LC used data from a single epoch at two channels, whereas TRAC used data from two epochs (of equal length) from a single channel. 

As suggested in [27], calculating LC from NIRS data required specific adaptations (see Equations (1)–(3) in [27]). Nevertheless, several problematic scenarios persisted with the proposed formulas. For instance, using Equation (1) from [27] could result in index insensitivity to response polarity; Equation (2) might saturate at extreme values of ±1 regardless of actual response amplitude, when hemispheric responses exhibit opposite signs; and Equation (3) might result in mathematical singularities if the denominator approaches or equals zero, particularly in the case of opposite responses. To address these issues, the authors of [27] explored a strategy of zeroing negative GLM coefficients, though this method could result in information loss. We proposed using the original normalization for LC, and, consequently, TRAC calculation. Our approach retained response polarity and did not require coefficient zeroing. When applying Equation (2) from [27] to our data, 31% of LC values reached 1 or −1, whereas only 5% of the index absolute values, calculated using our formula, exceeded 0.99. Using Equation (3) from [27] resulted in unbounded index values. For our data, the 95th percentile of the absolute LC values calculated using this equation was 6.63, with a maximum exceeding 6850. In contrast, our index remained bounded between −1 and 1. Therefore, our proposed formulas were less prone to the aforementioned issues, except in the rare case of zero response to all tasks, in which both indices were considered to be zero.

### 3.2. Laterality and Task Response Asymmetry in Patients

The HbO LC values, sequentially averaged over all sessions for each patient, then across all patients, and sequentially across all channels in the contralateral hemisphere, were 0.03 ± 0.22 during the imagery of the affected hand movement, and 0.10 ± 0.23 during the imagery of the intact hand movement (Figure 1a). The corresponding values for HbR LC were −0.01 ± 0.30 and −0.18 ± 0.27, respectively (Figure 1b).

The ART ANOVA analysis revealed a significant effect of Hand for HbR LC values (Table 2), indicating higher HbR LC values during affected hand movement imagery compared to intact hand movement imagery in patients. No other significant effects were found for HbO LC and HbR LC. Wilcoxon tests identified no channel pairs with significant HbO/HbR LC differences during contralateral hand movement imagery.

The values of HbO TRAC, sequentially averaged over all sessions for each patient, across all patients, and then across all channels for each hemisphere, were 0.18 ± 0.19 for the lesioned and −0.05 ± 0.20 for the intact hemisphere (Figure 1c). The corresponding HbR TRAC values were −0.15 ± 0.27 for the lesioned and 0.04 ± 0.23 for the intact hemisphere (Figure 1d).

ART ANOVA analysis revealed a significant effect of Hemisphere for HbO TRAC and HbR TRAC (Table 2), confirming significantly higher HbO TRAC and lower HbR TRAC in the lesioned versus intact hemisphere. Although no factor interactions reached significance, Wilcoxon tests identified specific channel pairs within the lesioned hemisphere that exhibited significant differences in HbO TRAC (Figure 1c).

### 3.3. Laterality and Task Response Asymmetry in Healthy Participants

The values of HbO LC, averaged across all channels in the contralateral hemisphere, were 0.14 ± 0.27 during the left-hand movement imagery and 0.16 ± 0.27 during the right-hand movement imagery (Figure 2a). The corresponding HbR LC values were 0.17 ± 0.34 and −0.18 ± 0.31, respectively (Figure 2b).

### 3.4. LC and TRAC Daily Dynamics

When daily LC and TRAC values, averaged across subjects, were pooled together by patient group, regression analysis revealed no significant time-dependent trends (FDR-adjusted *p* > 0.05 for all slopes). The same result was observed for the healthy group.

However, a more detailed examination of individual patient characteristics identified four participants, S1, S11, S14, and S15, who demonstrated substantial recovery of motor function. Among these, patients S1, S14, and S15 showed a day-by-day decrease in TRAC values within the lesioned hemisphere and a corresponding increase in the intact hemisphere (Figure 3). The dynamics reached significance in 10 channels (5 in the lesioned and 5 in the intact hemisphere) with 2 channels remaining significant after correction for multiple comparisons. In contrast, patient S11 displayed inverted dynamics, with increasing TRAC values in the lesioned hemisphere and decreasing values in the intact hemisphere (Figure 4). However, for S11, no channel dynamics remained significant after correction for multiple comparisons. Noteworthy, S11’s HbO and HbR responses did not differ from those of other patients in terms of the response sign and vividness. Similarly, the average TRAC value in S11 was more pronounced in the lesioned hemisphere, consistent with the overall patient pattern. 

No significant correlations were found between the slope coefficients and either the baseline motor function or the extent of improvement following interventions. Similarly, neither the day-averaged LC nor TRAC values showed significant associations with clinical outcomes.

## 4. Discussion

The analysis revealed that the HbR laterality coefficient in patients was significantly higher during the motor imagery of the paretic hand compared to the intact hand; however, the traditional index did not detect significant daily dynamics. In contrast, the proposed TRAC demonstrated greater sensitivity and clinical relevance: patients showed significantly higher HbO TRAC and significantly lower HbR TRAC in the affected hemisphere relative to the intact hemisphere. Among patients who achieved substantial motor function recovery, distinct daily fluctuations in HbO TRAC were observed, suggesting that this novel metric may serve as a more sensitive biomarker for tracking neuroplastic changes during rehabilitation. These findings indicated that TRAC provided superior discrimination of hemispheric differences and captured temporal dynamics overlooked by conventional laterality indices, potentially offering valuable insights into the mechanisms underlying post-stroke motor recovery.

Consistent with EEG and fMRI findings, studies examining metabolic brain activity with NIRS during actual or attempted movements in post-stroke patients have identified negative correlations between LC and motor deficit severity [26,31], as well as positive associations between LC increases and successful motor recovery [22,24,32]. In contrast, our study found no significant correlations between LC values and either baseline motor function or rates of motor function improvement. This discrepancy may be explained by our relatively small sample size and marked heterogeneity among patients, particularly in terms of time since stroke and initial motor deficit severity. Another contributing factor may be that motor imagery elicited weaker hemodynamic responses compared to actual motor execution [33], particularly in primary motor and sensory cortices [34,35]. In a study by Ramos-Murguialday et al. [11], where LC was calculated from fMRI data before and after EEG-BCI rehabilitation, activation shifts in the premotor and primary motor cortices toward the lesioned hemisphere correlated positively with motor recovery rates, but only when LC was derived from actual movement rather than motor imagery.

Therefore, the application of our proposed TRAC appeared especially beneficial for assessing interhemispheric asymmetry in post–stroke populations. TRAC quantified the differential involvement of specific brain regions during distinct motor tasks. Using this metric, we identified functional interhemispheric asymmetry during the motor imagery of both the affected and intact hands, even in the presence of significant group heterogeneity and the absence of meaningful LC temporal dynamics. The TRAC method revealed significantly larger differences in HbO and HbR concentration changes between affected and intact hand movement imagery in the lesioned hemisphere compared to the intact hemisphere. This pattern may reflect enhanced interhemispheric inhibition exerted by homotopic areas of the intact hemisphere [3], which could be more pronounced during intact hand movement tasks due to their hyperactivation. Interestingly, Stępień et al. [6] reported unaltered amplitude of event-related desynchronization in the lesioned hemisphere during intact hand finger movements while we observed decreased HbO response in the lesioned hemisphere during intact hand movement imagery.

Consistent with the concept that the intact hemisphere can assume some functions of the lesioned one [2,3,4,36], our analyses of HbR LC values in patients indicated that during intact hand movement imagery, more prominent responses occurred in the intact hemisphere, whereas during affected hand movement imagery, responses in both the intact and lesioned hemispheres were comparable. Thus, both metrics could be used complementarily.

In healthy individuals, no significant differences in HbO LC, HbR LC, or HbO TRAC values were observed when comparing left and right hemispheres or left- and right-hand movement imagery. This was consistent with existing evidence that unilateral motor tasks typically evoked stronger activation in contralateral motor areas with less pronounced activation in homologous ipsilateral regions–a pattern that was generally symmetric. However, ipsilateral activation increased with task complexity [37,38], especially in the left hemisphere [38], which may explain the observed trend toward lower absolute HbR TRAC values in the left hemisphere among healthy participants.

We acknowledge that the small sample size and high heterogeneity in the patient group were the primary limitations of our study. Nevertheless, these data reflect the typical conditions encountered in clinical BCI applications and are representative of the patient populations seen in real-world clinical practice. As previously discussed, these limitations may explain the absence of significant daily LC dynamics and the lack of correlations between asymmetry indices and motor function deficits or clinical outcomes. We believe that this limitation does not undermine our main conclusion regarding the potential of TRAC for assessing interhemispheric asymmetry based on fNIRS data.

As the current results did not establish associations between TRAC values–or their temporal dynamics–and baseline status or clinical outcomes, further research should focus on detailed investigation of TRAC trajectories and their relevance for motor function recovery. Additionally, evaluation of TRAC during actual motor execution, rather than imagery alone, warrants investigation. Moreover, the proposed metric extends beyond motor-related studies and could be applied to experiments involving various sensory or cognitive tasks.

## Figures and Tables

**Figure 1 sensors-25-05040-f001:**
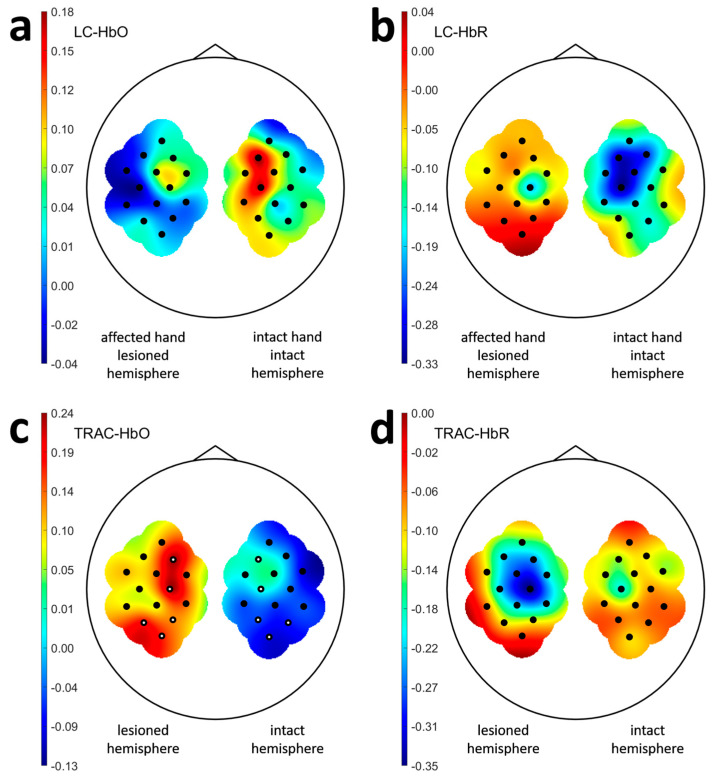
LC and TRAC values for the post-stroke subjects. The indices were averaged over each participant`s sessions and then over all the participants of the stroke group. Part (**a**) contains HbO LC values, (**b**) contains HbR LC values, (**c**) contains HbO TRAC values, and (**d**) contains HbR TRAC values. Topographic maps (**a**,**b**) show LC values only for hemispheres contralateral to the hands for which movements were imagined. Black dots indicate the channel positions, and white dots indicate the pairs of symmetric channels for which HbO TRAC values differed significantly according to Wilcoxon tests.

**Figure 2 sensors-25-05040-f002:**
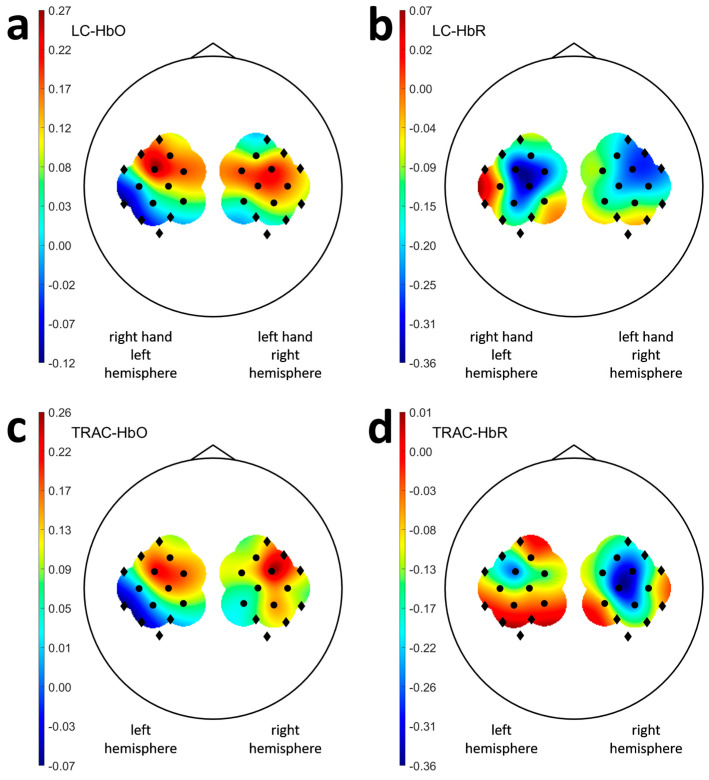
LC and TRAC values for the healthy subjects. The indices were averaged over each participant`s sessions and then over all the participants of the healthy group. Part (**a**) contains HbO LC values, (**b**) contains HbR LC values, (**c**) contains HbO TRAC values, and (**d**) contains HbR TRAC values. Topographic maps (**a**,**b**) show LC values only for hemispheres contralateral to the hands for which movements were imagined. Black dots indicate the channel positions, and black squares indicate the virtual positions of the channels used in the clinical experiments.

**Figure 3 sensors-25-05040-f003:**
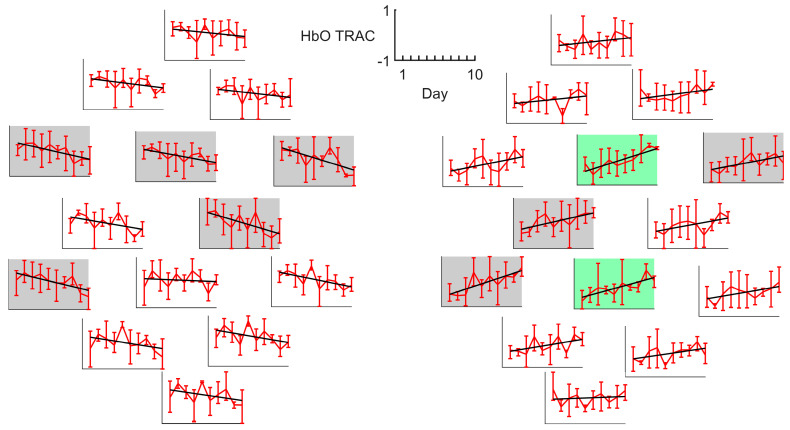
HbO TRAC dynamics for patients S1, S14, and S15, exhibiting a prominent rate of motor function recovery. Gray boxes indicate the channels with the significant dynamics (*p* < 0.05), while green boxes indicate the channels where the significance withstands correction for multiple comparisons. All axes have the same scale, with the *X*-axis corresponding to days and the *Y*-axis corresponding to the index values, as indicated by the reference frame in the middle.

**Figure 4 sensors-25-05040-f004:**
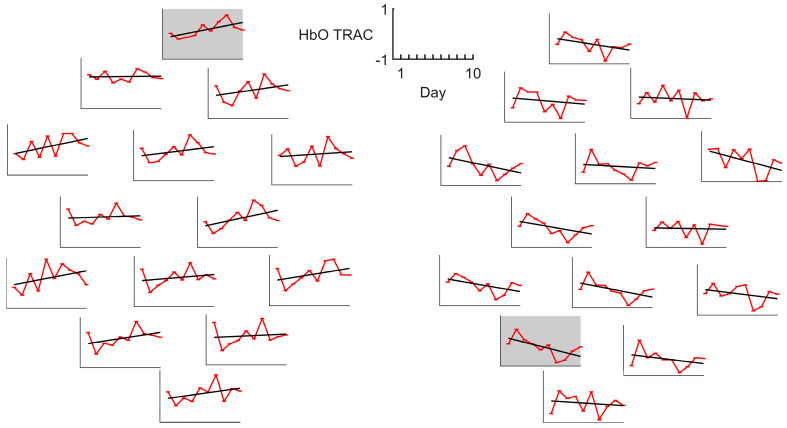
HbO TRAC dynamics for patient S11, who was among those exhibiting a substantial rate of motor function recovery. Gray boxes indicate the channels with the significant dynamics (*p* < 0.05). All axes have the same scale, with the *X*-axis corresponding to days and the *Y*-axis corresponding to the index values, as indicated by the reference frame in the middle.

**Table 1 sensors-25-05040-t001:** Patients’ characteristics.

ID	Sex	Age Range, y.o.	Stroke Time, Months	Lesioned Hemisphere	Baseline ARAT	ARAT Outcome	ARAT Improvement	TRAC Dynamics ^†^
S1	M	46–50	≤3	Right	35	55	57%	↘ **
S2	M	71–75	>6, ≤12	Left	44	50	14%	↘
S3	M	56–60	>6, ≤12	Right	35	41	17%	↗
S4	M	56–60	>6, ≤12	Left	39	43	10%	↘
S5	F	41–45	≤3	Left	52	57	10%	↗
S6	M	66–70	>6, ≤12	Left	1	1	0%	↗
S7	M	56–60	≤3	Right	49	57	16%	↘
S8	F	56–60	≤3	Left	38	45	18%	↗
S9	M	76–80	>12	Left	42	46	10%	↗
S10	F	56–60	>12	Left	10	10	0%	↘
S11	M	66–70	>3, ≤6	Right	6	16	167%	↗ *
S12	F	46–50	>3, ≤6	Left	24	29	21%	↗
S13	M	66–70	>6, ≤12	Right	46	50	9%	↗
S14	F	66–70	>6, ≤12	Right	4	9	125%	↘ **
S15	F	31–35	≤3	Right	19	28	47%	↘ **

ARAT—Action Research Arm Test; TRAC—task response asymmetry coefficient. † TRAC dynamics column indicates sign of average HbO TRAC slope in lesioned hemisphere (↘ stands for decrease, and ↗ stands for increase); * indicates presence of channels with significant dynamics, and ** indicates passed false discovery rate correction for some channels.

**Table 2 sensors-25-05040-t002:** Results of repeated-measures nonparametric factorial ART ANOVA.

Patients					Healthy				
**HbO LC**	**F**	**df**	**df.res**	***p*-value**	**HbO LC**	**F**	**df**	df.res	*p*-value
hand	1.81	1	392	0.18	hand	0.55	1	104	0.46
channel	0.11	13	392	0.99	channel	1.91	6	104	0.08
hand/channel	0.82	13	392	0.61	hand/channel	1.68	6	104	0.13
**HbR LC**	F	df	df.res	*p*-value	**HbR LC**	F	df	df.res	*p*-value
hand	28.49	1	392	<10^−6^	hand	0.38	1	104	0.54
channel	0.11	13	392	0.99	channel	1.73	6	104	0.12
hand/channel	1.13	13	392	0.32	hand/channel	1.14	6	104	0.35
**HbO TRAC**	F	df	df.res	*p*-value	**HbO TRAC**	F	df	df.res	*p*-value
hemisphere	34.03	1	392	<10^−6^	hemisphere	0.85	1	104	0.36
channel	0.32	13	392	0.99	channel	1.01	6	104	0.41
hemi/channel	0.33	13	392	0.99	hemi/channel	0.27	6	104	0.96
**HbR TRAC**	F	df	df.res	*p*-value	**HbR TRAC**	F	df	df.res	*p*-value
hemisphere	9.88	1	392	0.0018	hemisphere	0.82	1	104	0.06
channel	0.38	13	392	0.97	channel	0.98	6	104	0.44
hemi/channel	0.89	13	392	0.56	hemi/channel	0.26	6	104	0.73

HbO—oxyhemoglobin; HbR—deoxyhemoglobin; LC—laterality coefficient; TRAC—task response asymmetry coefficient.

## Data Availability

Data reported in this article are available on request from the corresponding author.

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
