# Peer review of "Hemodynamic Response Asymmetry During Motor Imagery in Stroke Patients: A Novel NIRS-BCI Assessment Approach"

_sensors, 2025, doi:10.3390/s25165040_

Round 1
Reviewer 1 Report
Comments and Suggestions for Authors
This manuscript presents a methodologically rigorous study introducing a novel Task Response Asymmetry Coefficient (TRAC) for quantifying interhemispheric hemodynamic asymmetry using fNIRS-BCI in stroke rehabilitation. There are some comments:
- The caption of Table 2 is “This is a table. Tables should be placed in the main text near to the first time they are cited. ” ????
- The titles of Section 3.2 and 3.3 are exactly the same.
- In Line 325, “patient S1 displayed …”, however, it is patient S11 in Fig. 4.
- In Fig. 3 and 4, Why did S11 show inverted trends?
- For sampling rate, 15.6Hz for patient participated while 3.9Hz for healthy subject. Why?
- In the captions of Fig.1 and 2, it said “(d) contains HbR LC values”. Please check it.
- In the results, only the data of 4 patients were discussed. Whether this discussion is statistically significant needs to be determined.
- In Line 342, it should be “TRAC”.
Author Response
We would like to thank you for your valuable feedback and apologize for technical issues which occurred during text transfer to the paper template. Following are the answers to your comments. The corresponding text changes are highlighted in yellow. Red font indicates English revision.
Comments 1. The caption of Table 2 is “This is a table. Tables should be placed in the main text near to the first time they are cited. ” ????
Response 1: The caption for Table 2 has been corrected. Now it is “Results of repeated measures nonparametric factorial ART ANOVA”, line 311 of the revised manuscript.
Comments 2. The titles of Section 3.2 and 3.3 are exactly the same.
Response 2: The titles of Sections 3.2 and 3.3 have been updated for clarity. The original Section 3.2 is now designated as Section 3.3 and is titled “Laterality and task response asymmetry in healthy participants”. The previous Section 3.3. is now 3.4 titled “LC and TRAC daily dynamics”, lines 325 and 340 of the revised manuscript.
Comments 3. In Line 325, “patient S1 displayed …”, however, it is patient S11 in Fig. 4.
Response 3: We appreciate the reviewer’s careful reading of our manuscript. The typo referring to “S1” has now been corrected to “S11”, line 350 of the revised manuscript.
Comments 4. In Fig. 3 and 4, Why did S11 show inverted trends?
Response 4: Upon re-examining the data from participant S11 we found no differences from the group averages reported in the paper with respect to response vividness, sign, or the average TRAC behavior. The average TRAC behavior in S11 was more pronounced in the lesioned hemisphere. These findings have been explicitly stated in the revised text, lines 353-356. Also, we updated the Discussion section to clarify that our results do not allow for definitive conclusion regarding the temporal TRAC dynamics or its association with clinical outcomes, lines 439-442 of the revised manuscript.
Thus, at this stage we are unable to give a conclusive answer to the question and consider this observation to require further research.
Comments 5. “For sampling rate, 15.6Hz for patient participated while 3.9Hz for healthy subject. Why?”
Response 5: The sampling rate of the NIRS system is determined by the illumination pattern, specifically the sequence in which the emitters are activated and number of emitters illuminated simultaneously. Originally, we used the default illumination pattern when each emitter is sequentially turned on. However, during our clinical experiments we established that the emitters could be grouped in four clusters to be activated simultaneously. This clustering approach allowed us to increase sampling rate. The revised manuscript now reflects this modification, lines 189-193.
Comments 6. “In the captions of Fig.1 and 2, it said “(d) contains HbR LC values”. Please check it.”
Response 6: The captions of Fig 1 and 2 were corrected, lines 297-303 and 332-338 of the revised manuscript.
Comments 7. Comments In the results, only the data of 4 patients were discussed. Whether this discussion is statistically significant needs to be determined.
Response 7: In the Results section, discussion of the subgroup of 4 patients is now confined to Section 3.4. Accordingly, the Section 3.4 has been revised, lines 341-356 of the manuscript. The cases demonstrating dynamic significance are now described in the text. Also, we have clarified in Section 3.2 that TRAC and LC averaging was performed across all patients, lines 291-292 and 315-316 of the revised manuscript.
Comments 8. In Line 342, it should be “TRAC”.
Response 8: Corrected, as suggested line 374 of the revised manuscript.
Sincerely,
Authors
Reviewer 2 Report
Comments and Suggestions for Authors
This study by Isaev et al introduces the novel metric TRAC to assess interhemispheric hemodynamic asymmetry in post-stroke patients using fNIRS during motor imagery-based brain-computer interface training. TRAC was shown to outperform the LC metric in detecting hemispheric differences and tracking daily neuroplastic changes. The paper is clear, well-written and provides compelling evidence that the TRAC approach could serve as a sensitive biomarker for monitoring stroke rehabilitation progress. However, I have some minor comments that I hope the authors will find useful.
Minor comments:
- TRAC rational and definition are only defined in Methods, whereas it is the highlight of the paper. Please move some of the relevant Methods (and even Introduction) materials into the new 1st sub-section of the Results section.
- For each 15 patients, am I understanding correctly that TRAC will have twice as much data as LC, because LC only considers 1 task? If so, what happens if you compute LC twice for each task and average the result?
- Figures 1 and 2 have different labels: Panels A & B (LC) use "right hand / left hand" as labels under the head. Panels C & D (TRAC) use "left hemisphere / right hemisphere". This switch in axis labeling conventions across panels makes it confusing to interpret and compare hemispheric responses. For clarity, I would suggest using consistent anatomical labels across all panels. Here is my suggestion : ‘Left hemisphere (right hand, lesioned) / Right hemisphere (left hand, intact)” for Figure 1 and ‘Left hemisphere (right hand, intact) / Right hemisphere (left hand, intact)’ for Figure 2.
- Figure 1 & 2: match the range of the color scale bar between A-C panels and B-D panels respectively. For example, in Figure 1, Panel A values range between -0.04-0.18 and panel C values range from -0.13-0.24.
- The caption states “Average LC and TRAC values for healthy participants”, but it’s unclear over which condition the averaging is performed (trials, days, participants…)
- Scientific format requires axis to have labels: color bars Figure 1 & 2 (LC-HbO, TRAC-Hb etc –the left and top headings can then be removed) and Figures 3 & 4 (no x or y axis label, and very small font size). I understand that adding all axis labels will unnecessarily clutter Figures 3 & 4, but please add at least one y-axis label.
- Title of Table 2 does not make sense
- Duplication of section titles: 2.5 and 2.6 have identical headings.
- Duplication of section titles: 3.2 and 3.3 have identical headings.
Author Response
We would like to thank you for your valuable feedback and apologize for technical issues which occurred during text transfer to the paper template. Following are the answers to your comments. The corresponding text changes are highlighted in yellow. Red font indicates English revision.
Comments 1. TRAC rational and definition are only defined in Methods, whereas it is the highlight of the paper. Please move some of the relevant Methods (and even Introduction) materials into the new 1st sub-section of the Results section.
Response 1: Relevant portions of the Methods and Introduction have been relocated to the newly created Section 3.1, lines 260-289 of the revised manuscript, where the properties of the proposed indices are discussed in detail. As no direct analogues for TRAC were identified in literature, the proposed normalization is compared with previously established approaches for LC.
Comments 2. For each 15 patients, am I understanding correctly that TRAC will have twice as much data as LC, because LC only considers 1 task? If so, what happens if you compute LC twice for each task and average the result?
Response 2: No, both indices use the same amount of data in terms of both the number of channels and samples per channel, or in terms of GLM coefficients, provided that each task is performed for an equal duration. The LC uses two symmetric channels and the same data epoch, while the TRAC index uses two data epochs and a single channel. In the case where a GLM is applied, the LC index uses two coefficients corresponding to a single task regressor at two different channels. TRAC index also uses two coefficients, one for the first task and the other for the second task, both coefficients derived from a single channel model.
This clarification is now reflected in Section 3.1 of the Results section, lines 268-271 of the revised manuscript.
Comments 3. Figures 1 and 2 have different labels: Panels A & B (LC) use "right hand / left hand" as labels under the head. Panels C & D (TRAC) use "left hemisphere / right hemisphere". This switch in axis labeling conventions across panels makes it confusing to interpret and compare hemispheric responses. For clarity, I would suggest using consistent anatomical labels across all panels. Here is my suggestion : ‘Left hemisphere (right hand, lesioned) / Right hemisphere (left hand, intact)” for Figure 1 and ‘Left hemisphere (right hand, intact) / Right hemisphere (left hand, intact)’ for Figure 2.
Response 3: The upper panels in Figures 1 and 2 combine the LC indices for two different conditions. The left side illustrates the LC in contralateral hemisphere during imagined movement of the affected (or left in case of the healthy subjects) hand, while the right side corresponds to LC in the contralateral hemisphere during imagined movement of the intact (or right) hand. This approach was chosen to avoid displaying LC values at symmetric channels, as these would simply be mirror images with opposite sign. We recognize that this presentation may lead to confusion and have corrected the figures and their captions, lines 296-303 and 331-338 of the revised manuscript.
Comments 4. Figure 1 & 2: match the range of the color scale bar between A-C panels and B-D panels respectively. For example, in Figure 1, Panel A values range between -0.04-0.18 and panel C values range from -0.13-0.24.
Response 4: The color scales were selected individually, as we did not compare TRAC and LC. We think that both indices are complementary and are not to be directly compared. Given that the main purpose of the figure is to illustrate interhemispheric asymmetry, we believe that preserving individuals color scales for each index most effectively serves this purpose.
Comments 5. The caption states “Average LC and TRAC values for healthy participants”, but it’s unclear over which condition the averaging is performed (trials, days, participants…)
Response 5: The captions were revised to clarify the averaging process. The statement “The indices were averaged over each participant`s sessions and then over all the participants of the stroke group” was added, lines 297-298 and332-333. Also, the averaging method was described to when the corresponding values were presented, lines 291-292 and 315-316.
Comments 6. Scientific format requires axis to have labels: color bars Figure 1 & 2 (LC-HbO, TRAC-Hb etc –the left and top headings can then be removed) and Figures 3 & 4 (no x or y axis label, and very small font size). I understand that adding all axis labels will unnecessarily clutter Figures 3 & 4, but please add at least one y-axis label.
Response 6: The figures have been updated, as suggested by adding a reference frame indicating the axes limits and displayed values, lines 358, 366. The figure captions have been revised accordingly by adding “All axes have the same scale with X-axis corresponding to days and Y-axis corresponding to the index values, as indicated by the reference frame in the middle”, lines 362-364 and 369-370 of the revised manuscript.
Comments 7. Title of Table 2 does not make sense.
Response 7: The caption for Table 2 have been corrected. It now reads: “Results of repeated measures nonparametric factorial ART ANOVA”, line 311 of the revised manuscript.
Comments 8-9. Duplication of section titles: 2.5 and 2.6 have identical headings.
Duplication of section titles: 3.2 and 3.3 have identical headings.
Response 8-9: The titles have been corrected, lines 216, 325, and 340 of the revised manuscript.
Sincerely,
Authors
Round 2
Reviewer 1 Report
Comments and Suggestions for Authors
I have no comments.